# Clinical Outcomes and Complication Rate after Single-Stage Hardware Removal and Total Hip Arthroplasty: A Matched-Pair Controlled Study

**DOI:** 10.3390/jcm12041666

**Published:** 2023-02-19

**Authors:** Francesco La Camera, Vincenzo de Matteo, Marco Di Maio, Raffaele Verrazzo, Guido Grappiolo, Mattia Loppini

**Affiliations:** 1IRCCS Humanitas Research Hospital, Via Alessandro Manzoni 56, 20089 Rozzano, Italy; 2Fondazione Livio Sciutto Ricerca Biomedica in Ortopedia-ONLUS, Via A. Magliotto 2, 17100 Savona, Italy; 3Section of Orthopaedic Surgery Department of Public Health, School of Medicine, “Federico II” University of Naples, via Sergio Pansini 5, 80131 Naples, Italy; 4Department of Biomedical Sciences, Humanitas University, Via Rita Levi Montalcini 4, 20090 Pieve Emanuele, Italy

**Keywords:** conversion THA, hardware removal, periprosthetic joint disease, one-stage

## Abstract

Background: Single-stage hardware removal and total hip arthroplasty is a complex surgical procedure, comparable to revision surgery. The purpose of the current study is to evaluate single-stage hardware removal and THA outcomes, compare this technique with a matched control group that has undergone primary THA and assess the risk of periprosthetic joint infection with a 24-month minimum follow-up. Methods: This study included all those cases treated with THA and concomitant hardware removal from 2008 to 2018. The control group was selected on a 1:1 ratio among patients who underwent THA for primary OA. The Harris Hip (HHS) and University of California at Los Angeles Activity (UCLA) scores, infection rate and early and delayed surgical complications were recorded. Results: One hundred and twenty-three consecutive patients (127 hips) were included, and the same number of patients was assigned to the control group. The final functional scores were comparable between the two groups; a longer operative time and transfusion rate were recorded in the study group. Finally, an increased incidence of overall complications was reported (13.8% versus 2.4%), but no cases of early or delayed infection were found. Conclusions: Single-stage hardware removal and THA is a safe and effective but technically demanding technique, with a higher incidence of overall complications, making it more similar to revision THA than to primary THA.

## 1. Introduction

Single-stage hardware removal and total hip arthroplasty (THA) is a viable surgical procedure following failed surgical treatment of a proximal femoral fracture, post-trauma end-stage osteoarthritis (OA) or following preserving surgery of the hip [1]. Conversion THA (cTHA) procedures have increased overtime because of the population aging associated with an increasing number of patients treated for hip fractures and the development of newer and safer hip-preserving surgical techniques [2].

cTHA with concomitant hardware removal may be a complex procedure with possible orthopedic and non-orthopedic complications, considering more similarities to revision THA in demographic, clinical, preoperative and early postoperative characteristics compared to primary THA [3,4]. The indications to perform cTHA concomitant to hardware removal are still debated [5]; single-stage cTHA has been demonstrated to be a viable and safe procedure [6], but some authors consider this surgery a risk factor for periprosthetic joint infections (PJI) due to the possible bacterial colonization of the hardware [7].

To our knowledge, only a few studies [5] have directly investigated septic complications rate following single-stage cTHA, and no clear guidelines or recommendations concerning infection screening prior to cTHA in patients with hardware in situ are available [8]. Remarkably, the last consensus meeting in 2018 did not provide a univocal answer concerning this topic [9,10] nor evidence to change perioperative antibiotic prophylaxis [11].

The purpose of this study was to investigate the clinical outcomes and complication rate in patients treated with single-stage hardware removal and THA compared with patients undergoing primary THA without hardware removal.

## 2. Materials and Methods

The present matched-pair study was based on medical records included in our institution’s registry of orthopedics surgical procedures. The study protocol for the development of this registry was approved by the Institutional Review Board (approval number 618/17). Written consent was given to store and process personal data and images for scientific purposes.

Patients treated in a high-volume tertiary referral center with single-stage hardware removal and THA from January 2008 to January 2018 were included in the study. The matched-pair control group was selected on a 1:1 ratio among patients who underwent THA for primary OA at the same institution and in the same period, with no hardware implant, using age, gender and length of FU (≥24 months) as matching criteria; none of the patients were lost to follow-up.

The selected inclusion criteria for the cTHA group were patients eligible for single-stage THA and hardware removal due to secondary hip arthritis and proper healing of the fracture or osteotomy site with a minimum of 2-year follow-up. The exclusion criteria were signs or symptoms suggestive of ongoing infection, history of previous infection in the index joint, elevated C-Reactive Protein (CRP) level (>1 mg/dL), recent (<6 month) intraarticular injection and pathological fractures. The preoperative joint aspiration was indicated for CRP and/or ESR (erythrocyte sedimentation rate) elevation or a high clinical suspicion for PJI due to multiple surgeries or a history of surgical site infection in the index joint or prior PJI. In the present study, no patient had preoperative joint aspiration in the synovial fluid leukocyte count and culture.

The preoperative assessment included a physical examination, laboratory tests including CRP and ESR and plain radiographs including an anterior–posterior (AP) view of the pelvis and axial view of the hip for THA. Bone scintigraphy, CT scan or MRI was performed according to surgeon preference. Preoperative CRP, demographical and surgical data are reported in Table 1.

The final clinical and radiographic evaluation and outcomes assessment were performed by two non-blinded independent authors not involved in the surgery with conventional radiographs in anteroposterior (AP) view of the pelvis and lateral view of the operated hip and patient-reported outcome (PRO) scores such as the Harris Hip Score (HHS) [12] and the University of California at Los Angeles Activity Score (UCLA) [13]. The HHS score ranged from 0 to 100 points, and it was classified as follows: excellent (between 90 and 100), good (80 to 89), fair (70 to 79) and poor (<70). The UCLA Activity Score ranged from 0 to 10, where 0 indicates “no physical activity, disability” and 10 “taking part to contact sports”, particularly indicated to evaluate a patient’s level of activity after implantation.

### 2.1. Surgical Procedure

Written informed consent was obtained from each patient before surgery. All surgeries were performed by five senior surgeons experienced in primary and revision hip and knee arthroplasty.

An accurate preoperative planning was carried out. The hardware was removed as the first surgical step, and implantation of the THA was conducted through a posterolateral approach and according to the femur first technique [14]. Blood-saving techniques were used in all cases. These included hypotensive locoregional anesthesia, side-lying position with the limb to be operated facing up, conscious sedation, heating devices and administration of tranexamic acid (10 mg/kg intraoperatively).

Implant choice was often oriented to a metaphyseal/diaphyseal fixation, hence making large use of “revision” stems (27.5% of cases); specifically, if a plate was present, it was generally indicated by a monoblock conical stem in order to manage the femur deformity [15]; in the cases of young patients with good bone stock, a standard or short-tapered stem was preferred; in case of a long nail, the choice fell back to a long revision monoblock or, more rarely, on a long modular stem; antibiotic-loaded bone cement was used for the cemented stem.

Indications of the index surgery, type of hardware removed and implants used are reported in Table 2.

In all cases, standard antibiotic prophylaxis was administered prior to the surgery (2 g cefazoline iv) and 1 g cefazoline every 8 h for 24 h post-surgery; in the case of a referred β-lactam antibiotics adverse reaction, vancomycin 1 g/12 h was administered; in the case of prolonged surgery time (over 120 min), an additional intraoperative dose of antibiotics was administered. No antibiotics were prescribed to the patients at discharge. At the time of the surgeries, there were no institutional protocols regarding tissue sampling; hence, according to surgeon preference, intraoperative samples were collected in 6 cases; there was no microorganism isolation.

### 2.2. Statistical Analysis

The statistical analysis was performed using STATA software version 15.

The continuous variables were firstly assessed by employing the Shapiro–Wilk test to assess the distribution of values, and then, the values were expressed by the average and standard deviation. For each variable, the comparison between the groups was analyzed with the Wilcoxon rank–sum test. The categorical variables were analyzed with the contingency table, as percentages, employing the Pearson chi-square test to assess the differences between the groups. A value *p* < 0.05 was considered significant.

## 3. Results

In the index period, 123 consecutive patients (127 hips) underwent single-stage hardware removal and THA. Eighty-seven patients had a cTHA for post-trauma OA (Figure 1 and Figure 2) and 36 patients (40 hips) for OA following preventive hip surgery (Figure 3); no patients were lost to follow-up. Cementless THA was preferred in all cases, except 4, in which a hybrid implant with a cemented stem and tantalum cup was used (Table 2). Based on the matching criteria, 123 patients were allocated to the control group. The hardware removal and control groups showed no significant differences in gender, age at surgery, body mass index (BMI), average FU length, preoperative Hb level and preoperative CRP level (Table 1).

The operative time was higher in the study group (108.9 min for cTHA, 58.3 min for primary THA) (Table 1). Postoperatively, a blood transfusion was required in 22 (17.9%) patients in the study group and 3 (2.4%) in the control group (*p* < 0.001).

At an average FU of 4.5 years, the HHS and UCLA scores showed no significant differences between the two groups and similar hospital lengths of stay.

According to the patient reports from the available clinical data, at the latest follow-up, there were registered neither early nor delayed PJI, and no radiographic loosening was observed according to Gruen et al. [16]. Five patients (4%) in the study group required further surgery: two cup revision and constrained liner insertion for recurrent dislocation; two stem revisions, one for postoperative subsidence and one for periprosthetic fracture; and one head exchange for a leg length discrepancy; there were also one intraoperative trochanteric fracture and three intraoperative calcar fractures registered. Five cases of early postoperative dislocation were registered in the study group, of which three were treated with a single closed reduction and two with surgical revision versus only one case in the control group. Intra- and postoperative complications were reported in Table 3.

The reported perioperative overall complications were higher in the study group. However, no nerve damage, pulmonary thromboembolism or deaths related to surgery occurred.

## 4. Discussion

The main finding of the present study was a comparable outcome between conversion THAs and primary THA in terms of similar hospital lengths of stay and similar HHS and UCLA scores. On the other hand, the conversion THAs reported a higher rate of complications, a consistently longer operative time and a greater likelihood of receiving a blood transfusion. In contrast to other authors [2], no deaths were recorded.

These findings are in agreement with Sierra et al. [17] and Ryan et al. [18], who previously reported a longer operative time, more consistent blood loss and a need for special surgical instrumentation in the conversion THAs. However, Schwartzkopf et al. [19] found longer LOS for patients that underwent conversion THA compared against primary THA and the employment of more hospital resources for these patients with significantly higher costs. Regarding implant characteristics, in most of the cases, a metaphyseal/diaphyseal fixation was needed, with a greater likelihood of requiring revision-type implant components and a need for special instrumentation [3,20]. However, conversion and primary THA procedures are currently bundled in the same DRG.

Moreover, our findings agree with Baghoolizadeh et. al. [3] that, from a perspective of change DRG classification of the cTHA, this surgery is more similar to revision THA than to primary THA in terms of demographic data and the complication rate. As mentioned before, a higher incidence of blood transfusion and overall peri-operative complications were reported (13.8% versus 2.4%); this result has to be related to a more difficult surgical procedure. Madariaga et al. [2] found even more cases that required a new surgery (23.6%).

In our study, no septic complications were reported after conversion THA. We could hypothesize that this finding resulted from an accurate selection of the sample that did not include cases of non-union and patients with preoperative CRP > 1 mg/dL. In our opinion, non-union is a highly suspicious condition for underlying infection that has to be extensively investigated and, due to this, is probably not suitable for a one-stage conversion. In previous studies, Madariaga et al. [2] found five cases of early PJI out of 55 patients (9.1%), and Scholten et al. [5] found a higher incidence in a single-stage group of early PJI compared to the two-stage group (8.6% versus 3.8%, *p* = 0.234). In both studies, a higher number of non-union cases were reported in the one-stage group. On the other hand, one previous study on 122 patients, including also cases of non-union (19%), and subjects with preoperative CRP > 1 mg/dL reported no PJIs after the index procedure [6]. Moreover, Mortazavi et al. [21] found a very low incidence of PJIs, reporting three cases out of 154 hips (0.51%), including also non-union cases.

Another interesting unintended finding concerns the role of preoperative hip aspiration. Given the lack of guidelines, no preoperative hip aspiration was performed. The hardware removal group came only from a selected cohort with no elevated CRP or any sign or history of infection [9] and a satisfactory bone healing; moreover, Madariaga et al. [2] performed a CT-guided aspiration in only seven cases on 55 and was negative in all cases, finding that no PJI were developed following microorganism isolation after THA conversion.

In our opinion, hip aspiration is not mandatory in a “hardware removal setting” with a proper bone healing of the site of fracture or osteotomy and no elevated CRP. To underpin this thesis, in the same setting, another high-volume center [6] performed prior-surgery aspiration only in 42% of cases and found just two positive cases, treated with the one-stage procedure and having no case PJIs; moreover, no strong correlation between hip aspiration and intraoperative cultures was noted [2].

This paper has, in our opinion, the following strengths: it has the longest FU on this issue and a high number of selected consecutive patients, with a matched control group; surgery was performed by the same high-volume team with the same standardized technique and using same indications and implants; lastly, it includes the highest number of DDH in the literature.

The present study has several limitations. This is a retrospective study, and the indication to perform preoperative aspiration or intraoperative cultures changed over the study period. Since cultures were not obtained in all patients, patients with low-grade infections could not be excluded [22]; however, none of the patients in the study was reoperated on for a periprosthetic infection. However, we included a homogeneous cohort of patients with a healed fracture or osteotomy with no elevated preoperative ESR and CRP. According to the most recent guidelines (2018 International Consensus Meeting on Musculoskeletal Infection), there is no indication to perform preoperative hip aspiration nor intraoperative sample cultures in these patients with no elevated preoperative ESR and CRP [23]. The study was conducted in a single center, and there was no randomized selection of patients. The time from index surgery was not reported. A low number of periacetabular fractures was included, and a very low number of intraoperative cultures was performed. Other variables (e.g., immunodeficiency, diabetes or malignancy) that affected the results were not reported.

## 5. Conclusions

Single-stage cTHA with concomitant hardware removal has demonstrated to be a safe and effective technique but, from a demographic and surgical complexity point of view, must be considered more similar to revision THA than primary THA. The success of this procedure is closely related to the proper selection of the patient, the careful choice of the implants and the experience of the surgeon. Overall, the risk of complications was higher, but an increased risk of early or delayed PJIs was not shown.

Further studies are needed to design viable guidelines in a “hardware removal setting” with elevated CRP and suboptimal healing of the site of osteosynthesis.

## Figures and Tables

**Figure 1 jcm-12-01666-f001:**
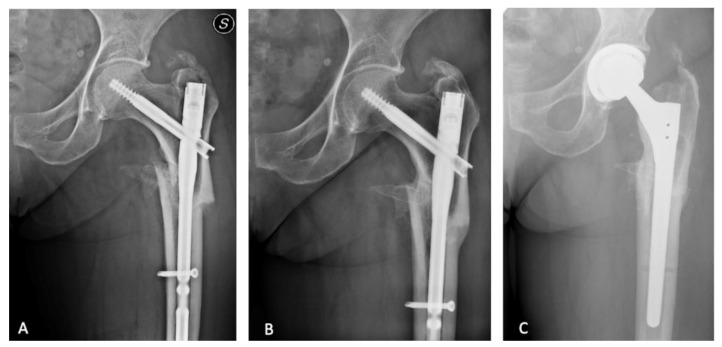
(**A**) Left pertrochanteric hip fracture treated with proximal femoral nail. S means left. (**B**) Fracture healed with varus and torsional deformity. (**C**) Left uncemented total hip arthroplasty with long revision conical stem.

**Figure 2 jcm-12-01666-f002:**
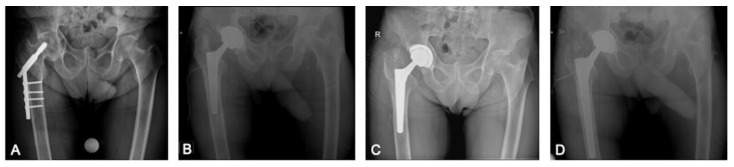
(**A**) Secondary hip osteoarthritis. (**B**) Postoperative Xray after conversion total hip arthroplasty with monoblock conical stem. (**C**) Stem subsidence at 3 days postoperatively treated with stem revision. R means right. (**D**) Early revision with long revision conical stem.

**Figure 3 jcm-12-01666-f003:**
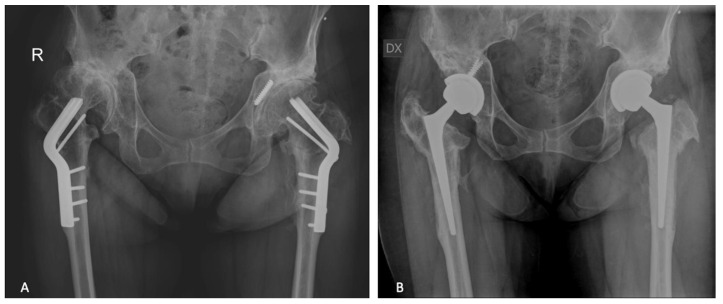
(**A**) Bilateral hip osteoarthritis after bilateral hip osteotomy. R means right. (**B**) Staged bilateral total hip arthroplasty combined with monoblock conical stem. DX means right.

**Table 1 jcm-12-01666-t001:** Demographic data.

Variables	Hardware Removal Group	Control Group	*p*-Value
Gender, n (%)			
Male	47 (38.2%)	47 (38.2%)	0.923
Female	76 (61.8%)	76 (61.8%)	
Age at surgery	61.9 ± 2.9 (16.6–94.4)	63.9 ± 2.9 (34.4–87.4)	0.24
BMI	27.9 ± 1.1 (15.3–43.3)	27.1 ± 1.1 (17.1–42.3)	0.64
FU	4.5 ± 0.3 years (2–11.4)	4.5 ± 0.5 years (2–11.4)	0.89
Surgery duration	108.9 ± 20.2 min (53–226)	58.3 ± 16.6 min (34–90)	<0.001
Preoperative Hb level	13.5 ± 0.31 mmol/L (1.7–5.9)	13.9 ± 0.25 mmol/L (1.9–6.1)	0.064
Preoperative CRP	0.381 ± 0.11 mg/dL (0.01–0.97)	0.471 ± 0.17 mg/dL (0.02–0.98)	0.38
LoS	5.26 ± 1.59 (3–12)	5.13 ± 1.6 (3–14)	0.36
Final HHS	90.2 ± 15.6 (38–100)	91.4 ± 13.05 (40–100)	0.51
Final UCLA Activity Score	5.72 ± 1.83 (1–9)	5.9 ± 1.79 (2–9)	0.43

BMI: Body Mass Index; FU: follow-up; Hb: hemoglobin; CRP: C-reactive protein; LoS: length of stay; HHS: Harris Hip Score; UCLA: University of California at Los Angeles. Age, BMI, FU, Surgery duration, Preoperative Hb level, CRP, LoS, HHS and UCLA Activity Score are expressed as the mean ± SD (range).

**Table 2 jcm-12-01666-t002:** Pathological and implant features.

	Post-Traumatic Group	Post-Preventive Group
Number of hips	87	40
Index surgery		
Subtrochanteric fracture	34 (39%)	-
Pertrochanteric fracture	45 (51%)	-
Shaft fracture	7 (8%)	-
Acetabular fracture	1 (1%)	-
Hip dysplasia	-	23 (57%)
Epiphysiolysis	-	14 (35%)
Osteonecrosis	-	1 (2%)
Arthrodesis	-	2 (5%)
Hardware removed		
Intramedullary nail	31 (24%)	-
Küntscher nail	1 (1%)	-
Dynamic hip screw	14 (11%)	-
Screws	35 (27%)	13 (10%)
plate and screws	6 (5%)	9 (7%)
Angled blade plate	-	12 (10%)
Tantalum rod	-	1 (1%)
K wires	-	1 (1%)
Camber	-	4 (3%)
Type of femoral stem used for conversion THA		
Cementless		
Conventional tapered	31 (24%)	7 (6%)
Short tapered	16 (13%)	8 (6%)
Conical	4 (3%)	21 (17%)
Long revision	31 (24%)	4 (3%)
Resurfacing	1 (1%)	-
Cemented		
Conventional tapered	4 (3%)	-
Surface-bearing		
Cer on poly	70 (55%)	28 (22%)
Met on poly	2 (1%)	-
Met on met	8 (6%)	-
Cer on cer	7 (5%)	12 (9%)

THA: total hip arthroplasty; Cer: ceramic; Poly: polyethylene; Met: metal.

**Table 3 jcm-12-01666-t003:** Perioperative notable events.

Event	Hardware Removal Group (N: 123)	Control Group (N: 123)	*p*-Value
Intraoperative calcar fracture	3	1	0.622
Postoperative periprosthetic fracture	1	0	1.000
Early postoperative dislocation	5	1	0.213
Intraoperative trochanteric fracture	1	0	1.000
Leg length discrepancy	1	1	1.000
Early postoperative stem subsidence	1	0	1.000
Reoperation/revision	5	0	0.060
Overall complications	17	3	0.002
Blood transfusion	22	3	0.001

## Data Availability

Data generated or analyzed during this study are included in this published article. Supplementary datasets used and/or analyzed during the current study are available from the corresponding author on reasonable request.

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
