# Peer review of "Clinical Outcomes and Complication Rate after Single-Stage Hardware Removal and Total Hip Arthroplasty: A Matched-Pair Controlled Study"

_jcm, 2023, doi:10.3390/jcm12041666_

Round 1
Reviewer 1 Report
The theme of the article is very interesting, written by the propositions of the JCM.
The title is clear and reflects well the main topic of the entire article. Introduction of the article represents a well written overview of single stage hardware removal and THA. This manuscript has a good educational value in total hip arthroplasty surgery. The discussion is well structured and explains the problems and dilemas.
Figures are clear.
Questions and comments:
1. Did the surgeons gave an antibiotic prophylaxis when surgery lasted over 120 minutes?
2. Criteria for detecting PJI is insufficient, especially for “low grade” PJI. It is obligatory for conducting 5 tissue cultures, sonication of explanted implants, synovial fluid leukocyte count for the diagnosis of PJI using Musculoskeletal Infection Society (MSIS) consensus criteria. Values of CRP and/ or ESR, clinical suspicion for PJI is not enough to detect PJI.
3. Did any of patients got perioperative tranexamic acid to decrease blood loose and consequently blood transfusion?
4. Hip aspiration during surgery is obligatory if we work by Musculoskeletal Infection Society (MSIS) consensus, in this way we increase sensitivity of our samples.
5. There were no randomize selection of patients in the study group and in the control group. In that way we have two identical groups to compare but researchers could potentially be bias when choosing patients for study.
6. We don’t know who examined and follow the patients after the surgery?
7. Did patients had CRP, ESR and X ray at last follow up?
Author Response
Thank you for your comments and suggestions which add value to our work.

Reviewer 2 Report
Major Comments:
1. More description should be provided on the number of surgeons for the patient's in the study at the tertiary referral center.
2. Since a single center was used, this should be listed as a limitation of this study.
3. Other variables should be included in Table 1, which affect outcomes. For example, patient characteristics should include: immunodeficiency or receiving immunosuppression; diabetes; malignancy; and others.
4. More details should be provided on what, if any, antibiotics were given once the patient was discharged and for how long.
5. A statement what and whether antibiotic cement/irrigation and/or any other antiseptic solution was used during surgery should be made.
6. A description should be provided on what matching criteria were used for controls.
7. Table 3 should include statistical analyses such as the overall complications and blood transfusion between the two groups.
8. No cultures obtained as a potential complication outcome is a limitation that should be acknowledged in the Discussion.
Minor Comments:
1. Please review the manuscript again for grammar and typos.
Author Response
Thank you for your comments and suggestions which add value to our work
